# Protective Effect of Knee Postoperative Fluid on Oxidative-Induced Damage in Human Knee Articular Chondrocytes

**DOI:** 10.3390/antiox13020188

**Published:** 2024-02-01

**Authors:** Roberta Giordo, Smitha Tulasigeri Totiger, Gianfilippo Caggiari, Annalisa Cossu, Andrea Fabio Manunta, Anna Maria Posadino, Gianfranco Pintus

**Affiliations:** 1Department of Biomedical Sciences, University of Sassari, Viale San Pietro 43/B, 07100 Sassari, Italy; rgiordo@uniss.it (R.G.); smithatulasigeri@gmail.com (S.T.T.); cossuannalisa@libero.it (A.C.); 2Orthopaedic and Traumatology Department, University Hospital, University of Sassari, Viale San Pietro 43/B, 07100 Sassari, Italy; gcaggiari@uniss.it (G.C.); andream@uniss.it (A.F.M.); 3Department of Medical Laboratory Sciences, College of Health Sciences, Sharjah Institute for Medical Research, University of Sharjah, Sharjah 27272, United Arab Emirates

**Keywords:** reactive oxygen species, osteoarthritis, chondrocytes, growth factors, post-operation knee fluid

## Abstract

The oxidative-stress-elicited deterioration of chondrocyte function is the initial stage of changes leading to the disruption of cartilage homeostasis. These changes entail a series of catabolic damages mediated by proinflammatory cytokines, MMPs, and aggrecanases, which increase ROS generation. Such uncontrolled ROS production, inadequately balanced by the cellular antioxidant capacity, eventually contributes to the development and progression of chondropathies. Several pieces of evidence show that different growth factors, single or combined, as well as anti-inflammatory cytokines and chemokines, can stimulate chondrogenesis and improve cartilage repair and regeneration. In this view, hypothesizing a potential growth-factor-associated action, we investigate the possible protective effect of post-operation knee fluid from patients undergoing prosthesis replacement surgery against ROS-induced damage on normal human knee articular chondrocytes (HKACs). To this end, HKACs were pre-treated with post-operation knee fluid and then exposed to H_2_O_2_ to mimic oxidative stress. Intracellular ROS levels were measured by using the molecular probe H_2_DCFDA; cytosolic and mitochondrial oxidative status were assessed by using HKACs infected with lentiviral particles harboring the redox-sensing green fluorescent protein (roGFP); and cell proliferation was determined by measuring the rate of DNA synthesis with BrdU incorporation. Moreover, superoxide dismutase (SOD), catalase, and glutathione levels from the cell lysates of treated cells were also measured. Postoperative peripheral blood sera from the same patients were used as controls. Our study shows that post-operation knee fluid can counteract H_2_O_2_-elicited oxidative stress by decreasing the intracellular ROS levels, preserving the cytosolic and mitochondrial redox status, maintaining the proliferation of oxidatively stressed HKACs, and upregulating chondrocyte antioxidant defense. Overall, our results support and propose an important effect of post-operation knee fluid substances in maintaining HKAC function by mediating cell antioxidative system upregulation and protecting cells from oxidative stress.

## 1. Introduction

Cartilage is a zonal, dense, aneural, avascular, and alymphatic connective tissue with excellent load-bearing functions, accounting for the easy mobility of joints [1]. Cartilage cells, called chondrocytes, are the primary mediators of the anabolic and catabolic processes that maintain extracellular matrix (ECM) integrity [2]. Chondrocytes respond to many factors, such as proinflammatory cytokines and growth factors, participating in ECM homeostasis by controlling the ECM’s degradation and synthesis [2]. The disruption or imbalance of this homeostasis leads to cartilage damage and breakdown, promoting degenerative joint diseases [3]. Among the potential risk factors for cartilage damage are age [4,5,6], biomechanical stress [7,8], sports injuries [9,10], the improper structure and mal-alignment of the joints [11,12], obesity [13], genetics [14,15], polymorphisms and mutations in extracellular matrix genes [16,17,18], and the excessive production of reactive oxygen species (ROS) [19,20,21]. Besides being involved in the oxidative metabolic turnover of cartilage tissue [22], ROS are also required to maintain essential intracellular signaling pathways and modulate ionic homeostasis, playing an important role in controlling chondrocyte function [23]. However, oxidative-stress-associated damage occurs when the cellular antioxidant capacity is insufficient to counteract excessive ROS production [20]. Elevated ROS levels may thus contribute to the degradation of cellular membranes and ECM components [24,25], the release of oxidized molecules in the cellular content, and chondrocytes’ senescence and death [21,26]. Such conditions increase synovial inflammation, which in turn increases ROS generation and the amount of degradation products [20]; indeed, synovium and activated chondrocytes are sources of proinflammatory cytokines, mainly IL-1beta, interferon-gamma, and TNF-alpha, which are involved in increasing ROS generation [27,28]. The association between this uncontrolled ROS production and the inappropriate cellular antioxidant defense is thus the primary trigger of osteoarthritis (OA) development and progression [29,30,31]. Though cartilage damage is the principal feature, OA is a whole joint disease involving the surrounding tissues (subchondral bone, ligaments, and synovium), and it is accompanied by changes in the subchondral bone and synovial inflammation [32]. Interestingly, OA has a non-systemic nature, having higher prevalence in large joints, such as the knee and hip; moreover, the main cytokine pool involved in its pathogenesis is derived from chondrocytes and synovial fibroblasts rather than from circulating blood inflammatory cells [33]. Therefore, in an effort to restore cartilage homeostasis, chondrocytes can also increase the levels of anti-inflammatory cytokines, such as IL-10, IL-4, and IL-13 [33,34]. Several pieces of evidence, both in vitro and in vivo, show that different growth factors, including transforming growth factor-β (TGF-β), insulin-like growth factor-I (IGF-I), and platelet-derived growth factor (PDGF), individually or combined, can stimulate chondrogenesis, improving cartilage repair and regeneration [35,36,37,38]. Accordingly, the application of growth factors may represent a promising treatment for cartilage defects and damage and for osteoarthritis treatment [38]. In this regard, in vitro studies have shown the importance of autologous human serum in chondrocyte monolayer expansion, which was shown to maintain chondrocyte proliferation and minimize chondrocyte apoptosis [39,40,41]. Moreover, human platelet lysate (HPL) has been identified as a valuable alternative to fetal bovine serum (FBS) to promote chondrocyte proliferation and maintain their chondrogenic features [42]. In this regard, platelet-rich plasma (PRP) therapy has gained prominence as a simple and non-invasive approach for knee arthritis treatment [43]. Indeed, by releasing anti-inflammatory cytokines, chemokines, and growth factors, PRP can stimulate chondrocyte proliferation and differentiation, ultimately reducing inflammation and promoting cartilage repair and regeneration [44]. The synovial membrane has a blood supply, and some soluble factors like inflammatory and anti-inflammatory cytokines can be present in the patient’s blood [45,46]. For instance, IL-6 and IL-10 concentrations were higher in patients with knee OA than in healthy controls [45]. In this view, considering a possible growth-factor-associated action, we hypothesized that post-operation knee fluid (POKF) from patients who underwent prosthesis replacement surgery would possess potential protective effects against oxidative-induced stress on human knee articular chondrocytes (HKACs). This study’s results support our hypothesis and suggest an important effect of POKF-contained substances in maintaining HKAC function by decreasing intracellular ROS levels, preserving cytosolic and mitochondrial redox homeostasis, and upregulating antioxidant cell defense, ultimately conserving the proliferation of oxidatively stressed HKACs.

## 2. Materials and Methods

### 2.1. Patients and Serum Sample Preparation

This study was conducted on 20 patients diagnosed with bilateral knee OA who underwent total knee arthroplasty using a posterior-stabilized implant (Nexgen LPs; Zimmer, Warsaw, IN, USA). Patients had a mean age of 70 years (range 58–74) and a male-to-female ratio of 2:3. Patients with coagulation problems, diabetes, liver diseases, or malnutrition were excluded from the study. None of the 20 patients received whole blood units through transfusion during the postoperative period. The post-operation knee fluid (POKF) and the post-operation peripheral blood sera (POPBS) were isolated as previously described [47]. POKF was centrifuged in Falcon tubes at 2200 rpm for 20 min at 37 °C; POPBS were instead treated in serum separator tubes and centrifuged at 2500× *g* for 10 min at 10 °C. The supernatant from both POKF and POPBS was then collected and stored at −80 °C.

### 2.2. Cell Culture and Treatments

Human knee articular chondrocytes (HKACs) were purchased from Innoprot (Catalog number #P10970, Derio (Bizkaia), Spain). HKACs were routinely grown in a chondrocyte medium with supplement (Innoprot #P60137) in a 5% (*v*/*v*) CO_2_ humidified atmosphere at 37 °C, as previously done for human endothelial cells [48]. On reaching confluence, the cells were passaged at a split ratio of 1:2 using trypsin–EDTA (Lonza, Basel, Switzerland) and subcultured until within passage five. Unless differently stated, HKACs were cultured in 96-well black plates (Corning, Lowell, MA, USA) until 70 to 80% sub-confluence. According to previous experimentation using human sera [49,50,51], cells were serum-starved for 8 h before being treated for 12 h with POKF and POPBS at a final concentration of 5%. For the ROS and redox status evaluations (cytoplasmic and mitochondrial), cells were treated with H_2_O_2_ at a 300 µM final concentration for 3 h, without removing the serum treatment.

### 2.3. Determination of Intracellular ROS Levels

The levels of intracellular ROS were determined using 2′,7′-dichlorodihydrofluoresceindiacetate (H_2_DCF-DA), as previously described, with minor modifications [51]. After 12 h of sera treatment, HKAC cells were incubated for 30 min in Hanks balanced salt solution (HBSS) containing 10 µM carboxy-H_2_DCFDA; cells were then washed with phosphate-buffered saline (PBS) and fluorescence was measured using a GENios plus microplate reader (Tecan, Männedorf, CH) at 485 nm excitation and 535 nm emission. Oxidative stress was induced by the addition of H_2_O_2_ at a sub-lethal dose of 300 µM [52].

The treatment-induced variation in fluorescence was kinetically measured over a time course of three hours. All fluorescence measurements were corrected for the background fluorescence and protein concentration. Results were evaluated by comparing measurements from five different experiments and expressed as the mean ± standard deviation (SD) of the relative fluorescence unit (RFU) values [53].

### 2.4. Determination of Cytosolic and Mitochondrial Redox Status

The cytosolic and mitochondrial redox status was measured using HKACs infected with lentiviral particles harboring the redox-sensitive green fluorescent protein (roGFP) [54,55]. The dynamic range of roGFP allows it to respond linearly to increasing doses of oxidants; moreover, roGFP can be targeted to different cellular compartments, including mitochondria, nuclei, and plasma membranes [54,55,56,57]. Cells stably transfected constitutively expressed roGFP in both cytosol and mitochondria, as observed under a fluorescence microscope. After 12 h of sera treatment, stable transfectants were washed with PBS and incubated with HBSS containing 300 uM H_2_O_2_ and only HBSS to serve as a control. The fluorescence was measured with a GENios plus microplate reader (Tecan, Männedorf, Switzerland), where the oxidized form of roGFP was read with a fluorescence excitation maximum of 400 nm, while the reduced form was read with a fluorescence excitation maximum of 485 nm. The fluorescence ratio between the readings at 400 nm and the readings at 485 nm gives information about the extent of the oxidative status in the cytosolic and mitochondrial compartments, respectively.

### 2.5. Determination of Cell Proliferation

Cell proliferation was measured via an ELISA-based assay for the quantification of 5-bromodeoxyuridine (The Cell Proliferation ELISA, BrdU, Chemiluminescent, Roche, Basel, Switzerland), incorporated into the genomic DNA of proliferating cells [58]. After 12 h of sera treatment, HKACs were incubated with DMEM with 2.5% FCS containing 300 µM H_2_O_2_ (or without H_2_O_2_ to serve as a control), plus the BrdU probe, at a 10 µM concentration for 10 h. After removing the supernatant, the cells were fixed with a Fix-Denat solution for 30 min. The Fix-Denat was then discarded, and cells were incubated with an anti-BrdU antibody conjugated to peroxidase (anti-BrdU-POD) for 90 min. Following three rinses with a washing buffer, a substrate solution was added and it was allowed to react for 3–10 min at room temperature. Finally, light emission was read using a GENios Plus microplate reader (Tecan, Männedorf, Switzerland). Results were normalized for protein content and expressed as the mean ± SD of the relative fluorescence unit (RFU) values.

### 2.6. Protein Extraction

After treatment, cells were washed with chilled PBS and incubated in ice-cold lysis buffer (CytoBuster; Novagen, Darmstadt, Germany) containing protease and phosphatase inhibitors for 10 min at 4 °C. Cells were then scraped, and the lysate was centrifuged at 16,000× *g* for 5 min at 4 °C. The supernatant was collected and stored at −80 °C. Protein content was determined using the Bradford assay following the manufacturer’s protocol (Sigma, St. Louis, MO, USA) [59].

### 2.7. Determination of Superoxide Dismutase (SOD) Activity

Superoxide dismutase (SOD) activity was determined using a superoxide dismutase (SOD) activity assay kit (BioVision, Abcam, Waltham, MA, USA) [60]. The kit utilizes a WST-1 molecule (Water-Soluble Tetrazolium 1) that generates a water-soluble formazan dye upon reduction by the superoxide anion, which is liberated following the addition of an enzyme solution present in the kit. The superoxide reduction rate is linearly correlated with the xanthine oxidase activity, and it is inhibited by SOD. SOD inhibition activity can be colorimetrically determined at 450 nm. Cells were treated with sera, and proteins were extracted and quantified as described in Section 2.6. The supernatant of each sample and different blanks containing equal protein amounts were used to measure the SOD activity. Samples and blanks were read using a plate reader at 450 nm (GENios Plus microplate reader, Tecan, Männedorf, Switzerland). To calculate the SOD activity (as inhibition rate %), the following equation was used:$$\mathrm{SOD}\;\mathrm{Activity}\;\left(\mathrm{inhibition}\;\mathrm{rate}\;\%\right)=\frac{\left(\mathrm{Ablank}1\;-\;\mathrm{Ablank}3)\;-\;(\mathrm{Asample}\;-\;\mathrm{Ablank}2\right)}{\left(\mathrm{Ablank}1\;-\;\mathrm{Ablank}3\right)}\times 100$$
where ABlank1 is the absorbance of the solution without the sample, ABlank2 is the absorbance of the solution without the enzyme working solution, and ABlank3 is the absorbance of the solution without the enzyme working solution and the sample.

### 2.8. Determination of Catalase Activity

Catalase is a ubiquitous antioxidant enzyme present in nearly all living organisms. It catalyzes the decomposition of hydrogen peroxide (H_2_O_2_) into water and oxygen. Here, catalase activity was determined using a fluorometric assay kit (BioVision, Abcam, Waltham, MA, USA) where catalase first reacts with H_2_O_2_ to produce water and oxygen; then, the unconverted H_2_O_2_ reacts with a probe (OxiRed™) to produce a fluorescent molecule, which can be measured at Ex/Em = 535/587 nm [61]. Catalase activity is reversely proportional to the signal. Cells were treated with sera, and proteins were extracted and quantified as described in Section 2.6. Fluorescence was quantified using a GENios plus microplate reader (Tecan, Männedorf, Switzerland) with an excitation wavelength of 535 nm and an emission wavelength of 590 nm [56]. The difference measured between the positive control’s and the sample’s fluorescence was used to determine the amount of H_2_O_2_ converted by the catalase present in the sample with respect to the H_2_O_2_ standard curve generated. Catalase activity can be calculated as follows:$$\mathrm{CAT}\;\mathrm{Activity}=\frac{B}{30*V}*\;\mathrm{Sample}\;\mathrm{Dilution}\;\mathrm{Factor}=\mathrm{nmol}/\min/\mathrm{mL}=\mathrm{mU}/\mathrm{mL}$$
where B is the decomposed H_2_O_2_ amount from the H_2_O_2_ standard curve (in nmol). V is the pre-treated sample volume added into the reaction well (in mL), and 30 is the reaction time (30 min). Unit definition: One unit of catalase is the amount of catalase that decomposes 1.0 μmol of H_2_O_2_ per min at pH 4.5 at 25 °C. Results were normalized for protein content and expressed as mU/mg.

### 2.9. Determination of Glutathione (GSH) Activity

The activity of glutathione (GSH) was measured by employing a fluorometric assay kit (BioVision, Abcam, Waltham, MA, USA) that provides a tool for the detection of GSH, GSSG, and total glutathione separately [62]. The O-phthalaldehyde (OPA) molecule reacts with GSH (not GSSG), generating fluorescence, thus specifically quantifying GSH. The addition of a reducing agent converts GSSG to GSH, so (GSH + GSSG) can be determined. To measure GSSG specifically, a GSH quencher is added to remove GSH, preventing reaction with OPA (while GSSG is unaffected). A reducing agent is then added to destroy the excess quencher and convert GSSG to GSH. Thus, GSSG can be specifically quantified. Cells were treated with sera, and proteins were extracted and quantified as described in Section 2.6. The GSH before and after adding the reducing agent was quantified using the GSH standard curve generated simultaneously in the 96-well black plate. Samples and standards were read using a fluorescence plate reader equipped with Ex/Em = 340/420 nm (GENios Plus microplate reader (Tecan, Männedorf, Switzerland)). Fluorescence values were corrected for background fluorescence and normalized for protein content, and the ratio of GSSG to GSH was used to determine the redox status of GSH in the cell.

### 2.10. Statistical Analysis

The results are displayed as the mean value along with the standard deviation (SD). To determine the statistical significance among various treatments, a one-way ANOVA followed by a post-hoc comparison Tukey test was performed. Wherever applicable, an unpaired *t*-test was used to compare the means of two sets of data. Any *p* values less than or equal to 0.05 were considered statistically significant.

## 3. Results and Discussion

### 3.1. Circulating Factors in POKF Protect Chondrocytes from H_2_O_2_-Induced Oxidative Stress

H_2_O_2_ is an important redox metabolite in redox sensing, signaling, and modulation [63]. Along with hydrogen sulfide (H_2_S) and nitric oxide (NO), H_2_O_2_ acts as a second messenger and activates (via specific oxidations) downstream pathways (such as homeostatic, pathological, or protective pathways), leading to different metabolic responses in cells, including cell proliferation, survival, or death [64,65]. In vitro studies have shown that H_2_O_2_ treatment can induce oxidative-stress-associated chondrocyte damage, producing genomic instability, reducing chondrocytes’ replicative ability, and inducing catabolic changes in cartilage composition by decreasing glycosaminoglycan (GAG) and proteoglycan synthesis [25,66,67,68]. Moreover, chondrocytes’ functions, such as DNA and protein synthesis, are negatively affected by H_2_O_2_-elicited ATP depletion [69]. We hypothesized that the circulating factors present in POKF may exert a protective effect against H_2_O_2_-induced oxidative damage in HKACs. We first investigated whether KOPF could counteract the H_2_O_2_-induced intracellular ROS increase to test our hypothesis. To this end, HKACs were pre-treated with 5% POKF and 5% POPBS (used as a control) for 12 h, respectively. Cells were then treated with the sub-lethal 300 µM H_2_O_2_ dose, as previously determined in viability dose–response experiments with H_2_O_2_ [47], and intracellular ROS levels were kinetically determined over a 3-h time course. Values at 2 h (steady state) were used for comparison (Figure 1A). As depicted in Figure 1A, both POKF- and POPBS-pre-treated HKACs maintained unchanged levels of ROS in the absence of H_2_O_2_. Compared to H_2_O_2_-untreated cells, a significant increase in ROS was observed in POPBS-pre-treated HKACs following H_2_O_2_ exposure (Figure 1A). In contrast, H_2_O_2_-exposed cells failed to show an ROS increase in POKF-pre-treated HKACs compared to H_2_O_2_-unexposed cells (Figure 1A). These results suggest that potential circulating antioxidant factors present in POKF, but not in POPBS, are able to counteract the H_2_O_2_-induced increase in ROS in cultured HKACs. The protective effect of POKF compared to POPBS is even more evident in Figure 1B, where the delta value (D), the difference in the intracellular ROS rise between the H_2_O_2_-treated and untreated groups, is significantly higher within the POPBS groups than the POKF groups. In summary, released factors in synovial fluid following prosthesis replacement surgery, rather than circulating blood factors, may act as antagonists of H_2_O_2_-induced oxidative stress by counteracting the increase in intercellular ROS.

### 3.2. Circulating Factors in POKF Preserve Cytosolic and Mitochondrial Redox Status in Oxidatively Stressed Chondrocytes

ROS are natural by-products of multiple enzymatic reactions in various cellular compartments, such as peroxisomes, the endoplasmic reticulum, mitochondria, and cytoplasm [70]. Cytoplasmic ROS are mainly produced by nicotinamide adenine dinucleotide phosphate (NADPH) oxidase (NOX) and nitric oxide synthase (NOS) activity and are involved in various metabolic processes, such as autophagy, glycolysis, oxidative phosphorylation, and the pentose phosphate pathway [70,71,72]. Mitochondria are the primary ROS source, mainly produced during the oxidative phosphorylation process [70,73]. ROS overproduction can mediate mitochondrial damage and contribute to a wide range of pathological conditions, including OA [71,74]. Mitochondrial dysfunction in OA chondrocytes is indeed associated with increased oxidative stress [75], mtDNA damage [76], increased mitochondrial membrane permeability, decreased activity of respiratory chain complexes II and III and ATP production, inflammatory responses, and cell death [77,78,79,80], which ultimately result in cartilage damage and degeneration [74]. Thus, we next investigated the ability of POKF to overcome H_2_O_2_-induced oxidative stress within specific cellular compartments, such as the mitochondria and cytosol. To this end, two human HKAC lines constitutionally expressing the redox-sensing green fluorescent protein (roGFP) in both cytosolic (cyto-roGFP) and mitochondrial (mito-roGFP) compartments were used to detect the cytosolic and mitochondrial redox status, respectively. Cyto-roGFP-HKAC and mito-roGFP-HKAC stable lines were pre-treated with 5% POKF and 5% POPBS for 12 h before exposure to the sub-lethal 300 µM H_2_O_2_ dose [47]. The mitochondrial and cytosolic redox states were then kinetically determined over a 3-h time course, and values at 2 h (steady state) were used for comparison (Figure 2A,B). In the absence of H_2_O_2_, the extent of redox status was nearly the same in both the cytosolic and mitochondrial compartments of HKACs pre-treated with both POKF and POPBS (Figure 2A,B). Upon treatment with H_2_O_2_, POKF pre-treatment was able to counteract the H_2_O_2_-induced oxidation in both compartments, while POPBS pre-treatment had no protective effect (Figure 2A,B). Indeed, non-significant changes in redox status were observed in both the cytosol (Figure 2A) and mitochondria (Figure 2B) of POKF-pre-treated HKACs exposed to H_2_O_2_, compared to the H_2_O_2_-unexposed HKACs. In contrast, POPBS pre-treatment failed to prevent oxidation increases, resulting in the significant elevation of cytosolic and mitochondrial oxidation compared to the H_2_O_2_-unexposed groups.

POKF’s protective effect against H_2_O_2_-induced redox changes is also highlighted by the low D value (small difference in redox changes between H_2_O_2_-treated and H_2_O_2_-untreated groups) in both the cytosolic (Figure 3A) and mitochondrial (Figure 3B) compartments of POKF-pre-treated HKACs as compared with POPBS-pre-treated HKACs. Indeed, a high D value (large difference in redox changes between H_2_O_2_-treated and H_2_O_2_-untreated groups), indicative of little or no protective effect, was found instead in both the cytosolic (Figure 3A) and mitochondrial (Figure 3B) compartments of POPBS-pre-treated HKACs as compared to POKF-pre-treated HKACs. The current findings indicate that the circulating factors present in the POKF can preserve both the cytosolic and mitochondrial redox status of HKACs under oxidative stress conditions. The protective effect exerted by POKF on the functionality of the two subcellular compartments may thus potentially contribute to preserving cell integrity and eventually reducing cartilage damage.

### 3.3. Circulating Factors in POKF Preserve the Proliferative Ability of Oxidatively Stressed Chondrocytes

Chondrocyte senescence is an age- and oxidative-stress-related factor inducing matrix homeostasis imbalance, affecting the cartilage repair efficacy, and contributing to OA development [81,82,83]. As oxidative stress was found to affect chondrocytes’ viability and inhibit their proliferation [83,84,85], we investigated the ability of POKFs to preserve or increase the proliferation of oxidatively stressed HKACs.

Under H_2_O_2_-induced oxidative conditions, POKF pre-treatment was able to preserve, but not increase, HKAC proliferation (Figure 4A). Indeed, POKF-pre-treated cells exposed to H_2_O_2_ showed the same proliferation levels as H_2_O_2_-unexposed cells. On the other hand, a significant decrease in HKAC proliferation was instead observed in POPBS-pre-treated HKACs, indicating the lack of protection against H_2_O_2_-elicited oxidative stress (Figure 4A). POKF’s ability to preserve HKAC proliferation is also indicated by the low D value (small difference in cell proliferation rate between H_2_O_2_-treated and untreated POKF groups) compared to the D value of the POPBS groups (Figure 4B). Indeed, the high D value (large difference in cell proliferation rate between H_2_O_2_-treated and untreated POPBS groups) is indicative of POPBS’s poor protective effect against the H_2_O_2_-induced HKAC proliferation decrement (Figure 4B). Besides corroborating the negative impact of oxidative stress on chondrocytes’ proliferation, these results also suggest that the growth factors released in the synovial fluid following prosthesis replacement surgery may preserve the proliferation of oxidatively stressed chondrocytes, possibly delaying the onset of senescence.

### 3.4. Soluble Factors in POKF Failed to Increase SOD Activity in Oxidatively Stressed Chondrocytes

Our current data show a protective role of POKF against the H_2_O_2_-induced (1) ROS increase, (2) cytosolic and mitochondrial redox imbalance, and (3) decreased chondrocyte proliferation. To further corroborate POKF’s antioxidant potential in oxidatively stressed HKACs, we investigated its ability to modulate the levels of relevant antioxidant enzymes such as superoxide dismutase (SOD), catalase, and glutathione (GSH). Superoxide dismutase (SOD) catalyzes the dismutation of the superoxide anion into hydrogen peroxide and molecular oxygen. There are three major families of SOD, depending on the protein fold and the metal cofactor: (1) the Cu/Zn type (which binds both copper and zinc), (2) the Fe and Mn types (which bind either iron or manganese), and (3) the Ni type (which binds nickel). Interestingly, chondrocytes constitutively express all these three enzymes; their expression has been reported to be high in normal cartilage and dramatically depleted in advanced OA cartilage, further exacerbating chondrocytes’ oxidative stress response and promoting their degeneration [86]. In this light, we believed that it was reasonable to check the level of SOD under our experimentally induced oxidative conditions. We evaluated the total SOD activity since the used protocol allowed us to determine both Cu/Zn SOD and Mn-SOD activity. Considering both the H_2_O_2_-exposed and H_2_O_2_-unexposed groups, chondrocytes pre-treated with POKF showed higher but not significant SOD activity compared to those pre-treated with POPBS (Figure 5).

Moreover, when comparing the H_2_O_2_-exposed groups with the H_2_O_2_-unexposed groups, both POKF and POPBS failed to significantly increase the SOD levels, suggesting that both treatments were unable to induce an increase in enzyme activity under the experimentally induced oxidative conditions. One reason for SOD’s limited role under our experimental conditions could be the presence of H_2_O_2_. Indeed, although H_2_O_2_ can elicit further ROS generation inside the cells, it is, however, itself a product of SOD, which can eventually modify the enzymatic activity. In fact, H_2_O_2_, if present in excess inside the cell, can induce modifications of the SOD structure determining the enzyme inhibition [87]. In this regard, a different scenario may be present in vivo with pathology-derived oxidative stress, where the POKF treatment might be able to increase the activity of SOD, thus providing antioxidant protection.

### 3.5. Soluble Factors in POKF Trigger Catalase Activity in Oxidatively Stressed Chondrocytes

Catalase is a ubiquitous antioxidant enzyme present in nearly all living organisms. It catalyzes the decomposition of H_2_O_2_ into water and oxygen, thus being an effective detoxifying enzyme. A recent study revealed that in patients with an age-related progressive degenerative joint disease, catalase is capable of counteracting hydrogen peroxide, protecting against chondrocyte injury and knee OA progression by suppressing oxidative stress [88]. This finding prompted us to investigate catalase activity under our experimental oxidative conditions. Contrary to POPBS, POKF pre-treatment increased the catalase levels in the H_2_O_2_-exposed group compared to the H_2_O_2_-unexposed one (Figure 6). Moreover, comparing the H_2_O_2_-exposed groups, the increase in catalase levels was significantly higher in POKF-exposed cells than in POPBS-exposed ones, suggesting a more potent antioxidant effect of POKF compared to POPBS (Figure 6). The data also indicated that POKF has potent antioxidant potential in normal conditions, as, when comparing H_2_O_2_-unexposed cells, POKF pre-treatment was able to significantly increase the catalase levels compared to POPBS (Figure 6).

### 3.6. Soluble Factors in POKF Lower GSSG/GSH Ratio in Oxidatively Stressed Chondrocytes

Glutathione is the major intracellular low-molecular-weight thiol, playing a critical role in tissue and cell defense against oxidative stress. Indeed, glutathione is essential in maintaining the proper cellular redox potential and protecting cells against oxidative damage. In the cells, glutathione can be either reduced (GSH) or oxidized (GSSG), and redox homeostasis and oxidative stress protection depend on the total glutathione concentration and the GSH/GSSG ratio [89]. In this context, it is also possible to express the relation between reduced and oxidized glutathione as the GSSG/GSH ratio, which is employed more often to disclose disturbances in the cellular/tissue redox metabolism [90]. The GSSG/GSH ratio is indeed commonly used as a biomarker of the redox balance in both cells and tissues, and its value is considered an index of cellular oxidative stress [90]. OA-related models of stressed cartilage disclose several mechanisms in which glutathione provides oxidative stress resistance and resilience, acting as a key player in modulating these phenomena [91]. Moreover, variations in chondrocytes’ oxidative status associated with cartilage changes during aging have been reported [19]. In fact, an age-induced increase in the GSSG/GSH ratio implies a higher basal level of oxidative stress associated with cartilage aging and diseases such as OA cartilage [92,93]. In view of this, we determined both reduced and oxidized glutathione in oxidatively stressed HKACs exposed to POKF and POPBS and expressed the value as the GSSG/GSH ratio. As depicted in Figure 7, no significant difference in the GSSG/GSH ratio was found between H_2_O_2_-exposed and H_2_O_2_-unexposed cells within the POKF group, and similar results were also reported for the POPBS group. However, comparing both the H_2_O_2_-exposed and H_2_O_2_-unexposed groups, cells pre-treated with POKF showed significantly lower GSSG/GSH ratios compared to the POPBS-treated ones, suggesting that the POKF treatment was able to trigger GSH synthesis, significantly minimizing oxidative stress. Our current findings demonstrate that POKF pre-treatment can increase the GSH concentration, maintaining GSH in its reduced form and ultimately decreasing the GSSG/GSH ratio. In this regard, it is important to note that a reduced cytosolic redox status is essential in maintaining the function of cellular proteins, enzymes, and redox-sensitive signaling pathways. Chondrocytes’ response to cytokines and growth factors is tightly linked to the cellular redox status and intracellular antioxidant systems’ action. Indeed, the involvement of ROS in mediating chondrocyte and ECM component damages has stimulated many studies on the role of endogenous and exogenous antioxidants as potential drugs to mitigate cartilage ROS damage [24,92,94]. Many of these have focused on the use of antioxidant medicinal plants [95], resveratrol [96], NSAIDs [97], and SOD mimetics [98], with the purpose of not only alleviating inflammation and pain but also locally protecting the cartilage against the harmful effects of ROS. Moreover, increasing the body’s antioxidant content through the diet [99], oral supplements [100], and/or their intra-articular administration [101] has also been suggested to combat ROS and decrease their damaging effects on chondrocytes [102]. However, clinical studies are required to determine the long-term effects of antioxidant diets, supplements, and possible drugs.

On the other hand, the potential antioxidant role of growth factors has not been widely explored yet, except in the study by Jallali, N. et al. [103], showing that IGF-1 was able to decrease ROS levels and ROS-mediated cell death and increase glutathione peroxidase (GPX) activity in rat articular cartilage. In this regard, in line with Jallali, N. et al., the results of our study support the potential use of endogenous growth factors as preventive and therapeutic approaches to combat the detrimental effects of ROS-induced cartilage damage. Our findings indeed indicate that POKF possesses the ability to preserve the growth of oxidatively stressed chondrocytes, protecting them from ROS-associated damage.

## 4. Conclusions and Future Directions

Our study showed that POKF could overcome exogenously induced oxidative stress and its associated inhibition of chondrocyte proliferation. The current findings support the hypothesis that growth factors in POKF play a crucial role in protecting the proliferation of HKACs against oxidative stress. Such an effect appears to be achieved through the upregulation of antioxidative enzymes, which effectively counteract the harmful effects of H_2_O_2_-induced oxidative stress, ultimately safeguarding HKAC proliferation. Based on the current data, we believe that POKF has the potential to serve as a valuable tool against oxidative-associated chondrocyte conditions and help to pave the way for new and innovative treatments. POKF, which is typically discarded, could be repurposed to treat chondropathies, early OA, or mild OA in other body regions by employing it in the patients from whom it was initially collected. Besides being safe, since it is obtained via an autologous donation, this approach could offer a minimally invasive and effective therapy that could help to alleviate the pain, inflammation, and stiffness associated with these conditions. Delivering POKF directly into the joint could target the underlying pathology and stimulate cartilage repair, leading to improved joint function and quality of life for affected patients. POKF’s therapeutic applications may also be envisaged in patients different from the fluid donors. Indeed, after appropriate separation and purification, the active fluid components can be therapeutically employed to treat patients suffering from chondropathies or OA. By using the fluid’s components, it may be possible to avoid the need for prosthetic implants, which is particularly beneficial for patients in the early stages of the disease. This approach may also be effective for patients with advanced chondropathies and/or OA who have not responded to other forms of treatment. This method of treatment has the potential to provide a more natural and effective way to manage these conditions, reducing the need for invasive procedures and improving patient outcomes. Given the results of our research, we believe that further exploration into the use of POKF could lead to the development of effective treatments for individuals suffering from cartilage damage caused by ROS.

## Figures and Tables

**Figure 1 antioxidants-13-00188-f001:**
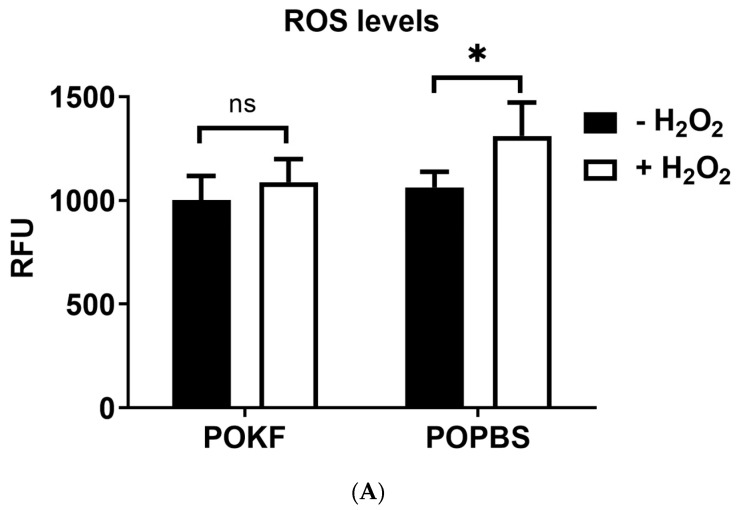
Effect of post-operation knee fluid and hematic sera on intracellular ROS levels in chondrocytes. (**A**) Human knee articular chondrocytes were pre-treated with the post-operation knee fluid (POKF) and the postoperative peripheral blood sera (POPBS) for 12 h before exposure to H_2_O_2_. ROS levels were assessed as described in Materials and Methods, 3 h after the POKF and POPBS treatments, in the absence or presence of H_2_O_2_. Results are expressed as relative fluorescence units (RFU). (**B**) The delta value (D) is the difference in RFU between H_2_O_2_-exposed and -unexposed cells in both POKF- and POPBS-pre-treated groups. *, significantly different from each other at *p* < 0.05; ns, not significantly different from each other at *p* < 0.05.

**Figure 2 antioxidants-13-00188-f002:**
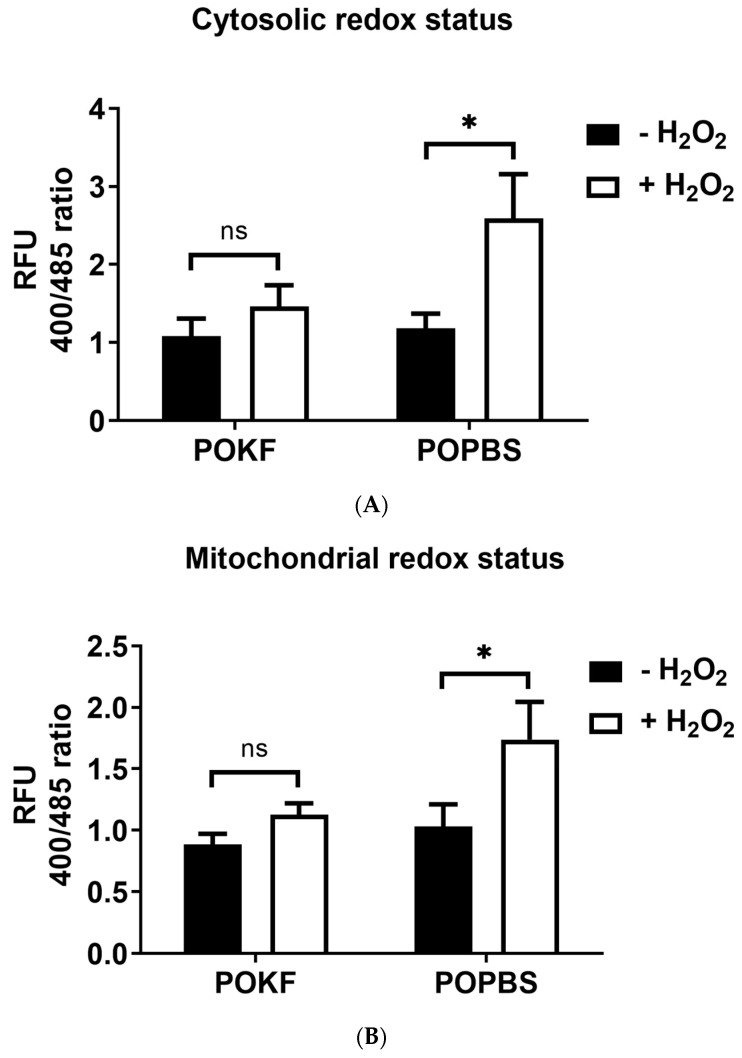
Effect of post-operation knee fluid and hematic sera on H_2_O_2_-induced changes in cytoplasmatic and mitochondrial redox status. Human knee articular chondrocytes were pre-treated with the post-operation knee fluid (POKF) and the postoperative peripheral blood sera (POPBS) for 12 h before exposure to H_2_O_2_. Cytoplasmic (**A**) and mitochondrial (**B**) redox status were assessed as described in Materials and Methods, 3 h after POKF and POPBS treatments, in the absence or the presence of H_2_O_2_. Results are expressed as the ratio of the relative fluorescence units (RFU). *, significantly different from each other at *p* < 0.05; ns, not significantly different from each other at *p* < 0.05.

**Figure 3 antioxidants-13-00188-f003:**
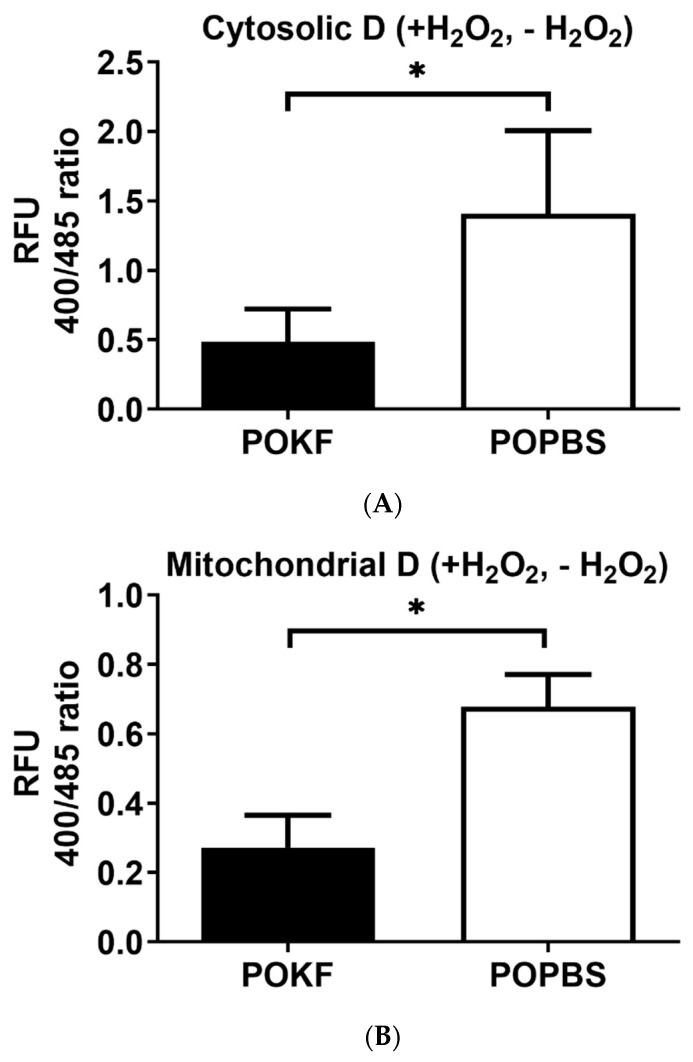
Changes in cytoplasmatic and mitochondrial redox status. The delta value (D) represents the difference in relative fluorescence units (RFU) between H_2_O_2_-exposed (+H_2_O_2_) and -unexposed (−H_2_O_2_) cells in both POKF- and POPBS-pre-treated groups for both cytosolic (**A**) and mitochondrial (**B**) roGFP-expressing HKACs. *, significantly different from each other at *p* < 0.05.

**Figure 4 antioxidants-13-00188-f004:**
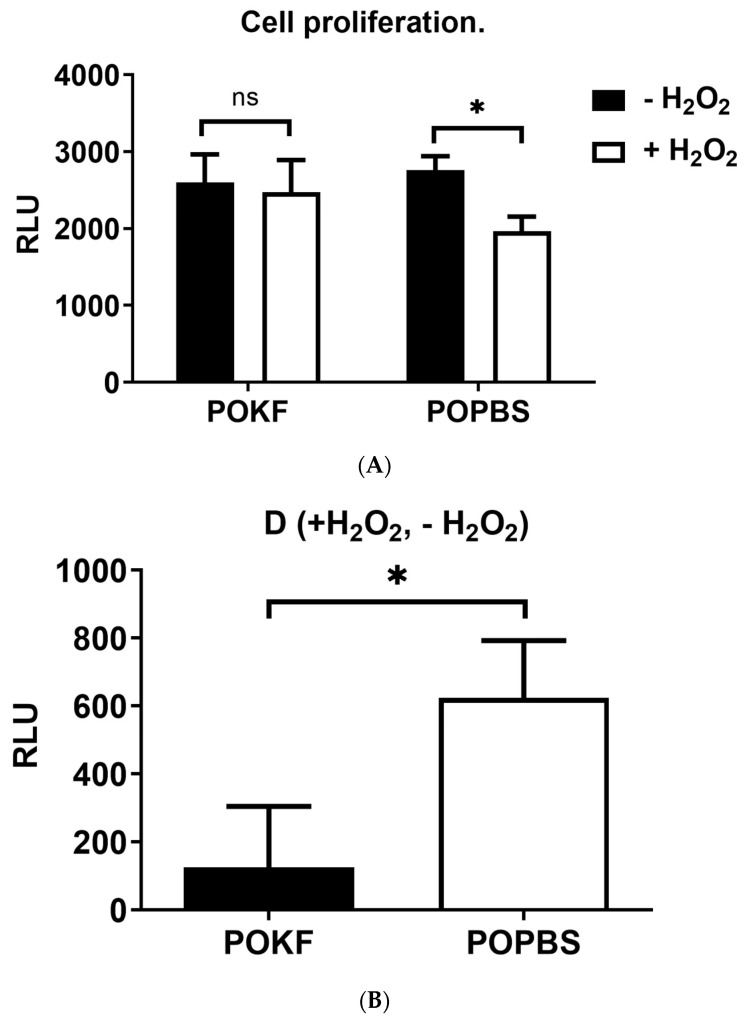
Effect of post-operation knee fluid and hematic sera on H_2_O_2_-induced changes in cell proliferation. (**A**) Human knee articular chondrocytes were pre-treated with the post-operation knee fluid (POKF) and the postoperative peripheral blood sera (POPBS) for 12 h before exposure to H_2_O_2_. Cell proliferation was evaluated as reported in Materials and Methods, 3 h after the POKF and POPBS treatments, in the absence (−H_2_O_2_) or presence (+H_2_O_2_) of H_2_O_2_. Results are expressed as relative luminescence units (RLU) (**B**). The delta value (D) is the difference in RLU between H_2_O_2_-exposed and -unexposed cells in both POKF- and POPBS-pre-treated groups. *, significantly different from each other at *p* < 0.05; ns, not significantly different from each other at *p* < 0.05.

**Figure 5 antioxidants-13-00188-f005:**
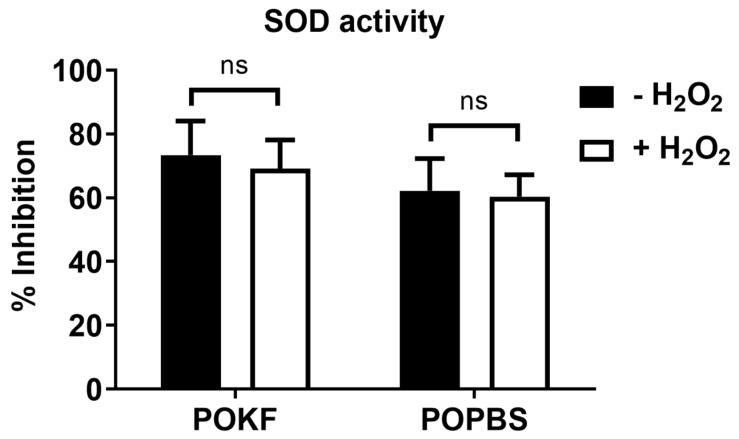
Effect of post-operation knee fluid and hematic sera on H_2_O_2_-induced changes in SOD activity. Human knee articular chondrocytes were pre-treated with the post-operation knee fluid (POKF) and the postoperative peripheral blood sera (POPBS) for 12 h before exposure to H_2_O_2_. SOD activity was evaluated as reported in Materials and Methods, 3 h after the POKF and POPBS treatments, in the absence or presence of H_2_O_2_. Results are represented as % of the inhibition activity of SOD. ns, not significantly different from each other at *p* < 0.05. SOD, superoxide dismutase.

**Figure 6 antioxidants-13-00188-f006:**
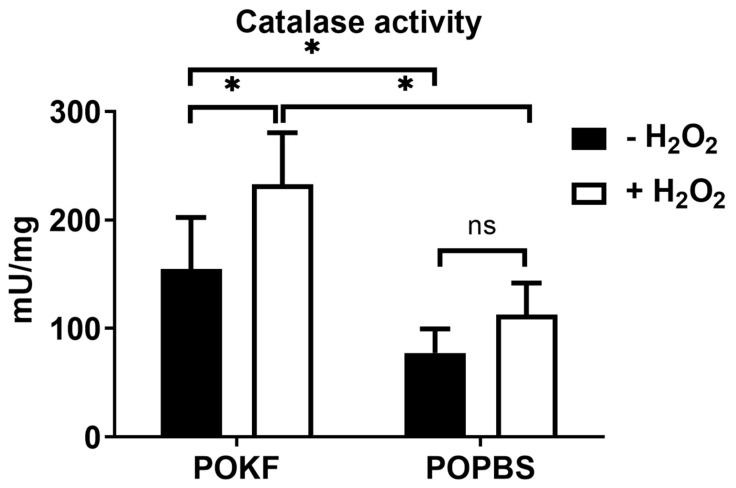
Effect of post-operation knee fluid and hematic sera on H_2_O_2_-induced changes in catalase activity. Human knee articular chondrocytes were pre-treated with the post-operation knee fluid (POKF) and the postoperative peripheral blood sera (POPBS) for 12 h before exposure to H_2_O_2_. Catalase activity was evaluated as reported in Materials and Methods, 3 h after the POKF and POPBS treatments, in the absence or presence of H_2_O_2_. Results are represented in mU/mg. *, significantly different from each other at *p* < 0.05; ns, not significantly different from each other at *p* < 0.05.

**Figure 7 antioxidants-13-00188-f007:**
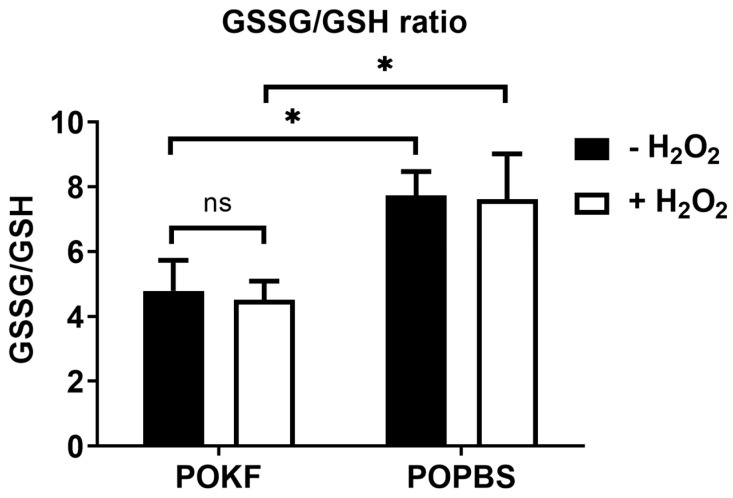
Effect of post-operation knee fluid and hematic sera on H_2_O_2_-induced changes in GSSG/GSH ratio. Human knee articular chondrocytes were pre-treated with the post-operation knee fluid (POKF) and the postoperative peripheral blood sera (POPBS) for 12 h before exposure to H_2_O_2_. GSSG/GSH ratio was evaluated as reported in Materials and Methods, 3 h after POKF and POPBS treatments, in the absence or presence of H_2_O_2_. Results are expressed as the ratio of the relative fluorescence units (RFU) obtained for both GSSG and GSH. *, significantly different from each other at *p* < 0.05; ns, not significantly different from each other at *p* < 0.05.

## Data Availability

All relevant data are available within the manuscript.

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
