# Peer review of "Protective Effect of Knee Postoperative Fluid on Oxidative-Induced Damage in Human Knee Articular Chondrocytes"

_antioxidants, 2024, doi:10.3390/antiox13020188_

Round 1

Reviewer 1 Report

Comments and Suggestions for Authors

In this study, Giordo et al. investigated the effect of post-operation knee fluid (POKF) on oxidative-induced damage in human knee articular chondrocytes. The authors found that POKF has a protective effect against oxidative-induced damage, such as ROS production (in both cytosolic and mitochondrial compartments) and chondrocyte proliferation. Additionally, the authors observed that POKF enhanced catalase activity. Furthermore, POKF treatment significantly decreased the levels of the GSSG/GSH ratio, implying the effect of POKF on minimizing oxidative stress. The manuscript is well-written. The introduction details the background and explains the rationale effectively. Results and discussion were carefully articulated. Although the responsible molecule(s) for the anti-oxidative effect in POKF were not narrowed down, the authors' attempt in this study was intriguing and will shed light on the biological effects of POKF, potentially leading to the development of a novel approach to treat chondropathies and OA. I have just a few comments about this manuscript.

In the figures, certain bars, such as those in Figure 5 and Figure 6 (POPBS), did not exhibit statistical significance. Please insert 'n.s.' (non-significant) between these bars. This adjustment will enhance the readability of the results.

Comments on the Quality of English Language

Minor English editing is recommended.

Author Response

We are delighted that our manuscript, entitled "Protective Effect of Knee Postoperative Fluid on Oxidative-Induced Damage in Human Knee Articular Chondrocytes," has been considered potentially suitable for publication in Antioxidants upon revision. We greatly appreciate the Reviewers' careful reading, balanced critiques, and constructive comments.

We have implemented most of the reviewers' suggestions, which have been highlighted in red within the resubmitted manuscript; we have also adjusted the manuscript in some places to better reflect the reviewers' requests.

Please find below a point-by-point response to the reviewers' comments and concerns.

In this study, Giordo et al. investigated the effect of post-operation knee fluid (POKF) on oxidative-induced damage in human knee articular chondrocytes. The authors found that POKF has a protective effect against oxidative-induced damage, such as ROS production (in both cytosolic and mitochondrial compartments) and chondrocyte proliferation. Additionally, the authors observed that POKF enhanced catalase activity. Furthermore, POKF treatment significantly decreased the levels of the GSSG/GSH ratio, implying the effect of POKF on minimizing oxidative stress. The manuscript is well-written. The introduction details the background and explains the rationale effectively. Results and discussion were carefully articulated. Although the responsible molecule(s) for the anti-oxidative effect in POKF were not narrowed down, the authors' attempt in this study was intriguing and will shed light on the biological effects of POKF, potentially leading to the development of a novel approach to treat chondropathies and OA. I have just a few comments about this manuscript.

In the figures, certain bars, such as those in Figure 5 and Figure 6 (POPBS), did not exhibit statistical significance. Please insert 'n.s.' (non-significant) between these bars. This adjustment will enhance the readability of the results.

Response: We thank the reviewer for this valuable suggestion. We have now modified the implicated figures as suggested by the reviewer.

Comments on the Quality of English Language. Minor English editing is recommended.

Response: As suggested by the reviewer, we have now carefully proofread the entire manuscript.

Reviewer 2 Report

Comments and Suggestions for Authors

The manuscript is well written and ist topic matches with the journal. In the beginning it is not clear to which purpose the fluid could be used in future. A potential future approach is mentioned at the end of the article and should be mentioned earlier. The fused result and discussion section does not allow a more global discussion e.g. of this future approach. The major issue ist hat the contents of the postoperative fluid are unknown. Was some analysis done by ELISA etc? This should be mentioned in the manuscript. Which particular growth factors etc. could exert antioxidant effects on chondrocytes? The time post  surgeries at which the postoperative fluid was taken and also the amount harvested from one patient should be mentioned (section 2.1.). Could the fluid be harvested several times post OP from the same patient?

Minor issues

Line 41: heterogenous, write „zonal“

Line 50: family history („genetic“)

Line 101: „bilateral“, why only bilateral?

Line 121 versus line 128: „hours“ versus „hrs“ please write it in a consistent manner, compare line 158 and the legend of figure 2

Line 136: „by comparing five measurement“ were the measurements conducted in independent experiments?

Line 151 and 153: „400nm“ insert blank „400 nm“

Legend of figure 1: add „in chondrocytes“

Line 295: bring the references in one shared bracket

Legend of figure 2: write the complate heading in bold

Legend of figure 3: add the abbreviations (+H2O2) and (-H2O2) tot he legend

Line 404: write „increase“

Line 405: write „to induce“

Legend of figure 5: explain „SOD“ in the legend

Line 480-481: here the envisaged approach could be better explained

Line 493: associated chondrocyte cell death: this was not analyzed

Comments on the Quality of English Language

The manuscript is well written and ist topic matches with the journal. In the beginning it is not clear to which purpose the fluid could be used in future. A potential future approach is mentioned at the end of the article and should be mentioned earlier. The fused result and discussion section does not allow a more global discussion e.g. of this future approach. The major issue ist hat the contents of the postoperative fluid are unknown. Was some analysis done by ELISA etc? This should be mentioned in the manuscript. Which particular growth factors etc. could exert antioxidant effects on chondrocytes? The time post  surgeries at which the postoperative fluid was taken and also the amount harvested from one patient should be mentioned (section 2.1.). Could the fluid be harvested several times post OP from the same patient?

Minor issues

Line 41: heterogenous, write „zonal“

Line 50: family history („genetic“)

Line 101: „bilateral“, why only bilateral?

Line 121 versus line 128: „hours“ versus „hrs“ please write it in a consistent manner, compare line 158 and the legend of figure 2

Line 136: „by comparing five measurement“ were the measurements conducted in independent experiments?

Line 151 and 153: „400nm“ insert blank „400 nm“

Legend of figure 1: add „in chondrocytes“

Line 295: bring the references in one shared bracket

Legend of figure 2: write the complate heading in bold

Legend of figure 3: add the abbreviations (+H2O2) and (-H2O2) tot he legend

Line 404: write „increase“

Line 405: write „to induce“

Legend of figure 5: explain „SOD“ in the legend

Line 480-481: here the envisaged approach could be better explained

Line 493: associated chondrocyte cell death: this was not analyzed

Author Response

We are delighted that our manuscript, entitled "Protective Effect of Knee Postoperative Fluid on Oxidative-Induced Damage in Human Knee Articular Chondrocytes," has been considered potentially suitable for publication in Antioxidants upon revision. We greatly appreciate the Reviewers' careful reading, balanced critiques, and constructive comments.

We have implemented most of the reviewers' suggestions, which have been highlighted in red within the resubmitted manuscript; we have also adjusted the manuscript in some places to better reflect the reviewers' requests.

Please find below a point-by-point response to the reviewers' comments and concerns.

The manuscript is well written and its topic matches with the journal. In the beginning it is not clear to which purpose the fluid could be used in future. A potential future approach is mentioned at the end of the article and should be mentioned earlier. The fused result and discussion section does not allow a more global discussion e.g. of this future approach.

Response: We thank the reviewer for this observation. We have now fixed these different aspects in the revised version of the manuscript.

The major issue is that the contents of the postoperative fluid are unknown. Was some analysis done by ELISA etc? This should be mentioned in the manuscript. Which particular growth factors etc. could exert antioxidant effects on chondrocytes?

Response: The evolution of osteoarthritis causes the progressive deterioration of the joint, ultimately requiring a joint prosthesis implant to obtain joint recovery in the disease's advanced stages. To facilitate recovery, a drainage system is put in place during surgery to prevent blood from collecting in the joint after the postoperative bleeding. Previous studies have identified plasma components in this liquid (e.g., anti-inflammatory cytokines) capable of promoting anti-inflammatory processes and tissue repair [1-3]. However, so far, no studies have identified the exact constituents of the fluid and the components present in it. In this regard, although usually discarded, our work aimed to evaluate the potential protective effect of the fluid in oxidatively stressed chondrocytes, which is the study's novelty. In this view, we did not analyze the postoperative fluid content. Nonetheless, taking into account the reviewer's valuable suggestion, among the future objectives of this study, we intend to investigate the fluid composition and analyze the effect of different components, including potential exosomes or microvesicles, in consideration of the promising results of preclinical studies using mesenchymal stem cell (MSC)-derived exosomes [4]

The time post surgeries at which the postoperative fluid was taken and also the amount harvested from one patient should be mentioned (section 2.1.). Could the fluid be harvested several times post OP from the same patient?

Response: In our study, we meticulously collected drainage fluid approximately 3 hours post-surgery. During this time frame, the quantity of fluid in the reservoir varied from patient to patient, ranging between 200 to 400 ml. To provide a representative analysis, we calculated an average volume across 20 patients, which resulted in an approximate mean volume of 270 ml.

Postoperatively, the fluid was capable of being harvested multiple times within a 48-hour window. This time limit was dictated by the duration for which the drainage system remained connected between the surgical site and the reservoir. Beyond this 48-hour period, the drainage apparatus was removed, precluding any further collection of drainage fluid.

A key focus of our study was the analysis of growth factors within the collected fluid. Based on our observations and current understanding of wound healing processes, we posit that a significant increase in growth factors predominantly occurs during the initial hours following surgery. Consequently, we hypothesize that the fluid harvested shortly after the operation is most likely to be enriched with these growth factors. This early postoperative period, therefore, represents a critical window for collecting fluid that may provide valuable insights into the dynamics of surgical recovery and the role of growth factors in this process.

Minor issues

Line 41: heterogenous, write „zonal“                        Done

Line 50: family history („genetic“)                Done

Line 101: „bilateral“, why only bilateral?

Response: In the context of bilateral osteoarthritis, we observe a distinct pathological pattern that is characteristically degenerative and affects both joints. This bilateral manifestation is pivotal in understanding the nature of degenerative joint diseases, particularly when contrasted with unilateral osteoarthritis. The latter often results from specific anomalies or pathologies affecting a single joint, such as cartilage deterioration, malalignment, maltracking, or other monoarticular conditions. These unilateral cases do not typically present the same pattern as idiopathic degenerative osteoarthritis, which is more uniformly distributed across both joints. Our study aims to dissect these differences further, providing a clearer understanding of the degenerative processes in bilateral versus unilateral osteoarthritis. This distinction is crucial for tailoring treatment strategies and understanding the progression of osteoarthritic conditions, as well as for the development of more effective therapeutic interventions targeted at the underlying mechanisms of these degenerative changes.

Line 121 versus line 128: „hours“ versus „hrs“ please write it in a consistent manner, compare line 158 and the legend of figure 2                         Done

Line 136: „by comparing five measurement“ were the measurements conducted in independent experiments? Yes. We have amended this statement in the revised version.

Line 151 and 153: „400nm“ insert blank „400 nm“             Done

Legend of figure 1: add „in chondrocytes“               Done

Line 295: bring the references in one shared bracket                                   Done

Legend of figure 2: write the complete heading in bold                   Done

Legend of figure 3: add the abbreviations (+H2O2) and (-H2O2) to the legend                 Done

Line 404: write „increase“                 Done

Line 405: write „to induce“               Done

Legend of figure 5: explain „SOD“ in the legend                 Done

Line 480-481: here the envisaged approach could be better explained

Response: As requested by the reviewer, we have rephrased this sentence to make it clearer

Line 493: associated chondrocyte cell death: this was not analyzed

Response: As correctly pointed out by the reviewer, cell death was modified with cell proliferation.

References

  1. Wang-Saegusa, A., et al., Infiltration of plasma rich in growth factors for osteoarthritis of the knee short-term effects on function and quality of life. Archives of orthopaedic and trauma surgery, 2011. 131: p. 311-317.
  2. Filardo, G., et al., Platelet-rich plasma intra-articular knee injections for the treatment of degenerative cartilage lesions and osteoarthritis. Knee Surgery, Sports Traumatology, Arthroscopy, 2011. 19: p. 528-535.
  3. Baltzer, A., et al., Autologous conditioned serum (Orthokine) is an effective treatment for knee osteoarthritis. Osteoarthritis and cartilage, 2009. 17(2): p. 152-160.
  4. Kim, Y.G., J. Choi, and K. Kim, Mesenchymal Stem Cell‐Derived Exosomes for Effective Cartilage tissue repair and treatment of osteoarthritis. Biotechnology journal, 2020. 15(12): p. 2000082.

Reviewer 3 Report

Comments and Suggestions for Authors

In this manuscript, the authors examined the protective effects of knee postoperative fluid on oxidative-induced damage in human knee articular chondrocytes. They found that the fluid inhibited oxidative stress as compared to post-operation peripheral blood sera. This manuscript is of interest from the point of elucidating the self-regulatory mechanism of oxidative stress resistance in cartilage tissue. However, there are some concerns, and they are discussed below.

1.       The authors added the fluid and measured oxidative stress level, redox status, and cell proliferation. The reviewer is curious whether the fluid has any effects on chondrocytic differentiation or not.

2.       The authors used the fluid and serum at the final concentration of 5% in cell culture experiments. The reviewer is curious how the authors decided the concentration?

3.       Also, the reviewer is curious in which ingredient(s) has inhibitory effect on oxidative stress. Did the author try to narrow down candidates such as heat denaturation against protein, and separation with centrifugal filtration against exosomes?

Author Response

We are delighted that our manuscript, entitled "Protective Effect of Knee Postoperative Fluid on Oxidative-Induced Damage in Human Knee Articular Chondrocytes," has been considered potentially suitable for publication in Antioxidants upon revision. We greatly appreciate the Reviewers' careful reading, balanced critiques, and constructive comments.

We have implemented most of the reviewers' suggestions, which have been highlighted in red within the resubmitted manuscript; we have also adjusted the manuscript in some places to better reflect the reviewers' requests.

Please find below a point-by-point response to the reviewer' comments and concerns.

In this manuscript, the authors examined the protective effects of knee postoperative fluid on oxidative-induced damage in human knee articular chondrocytes. They found that the fluid inhibited oxidative stress as compared to post-operation peripheral blood sera. This manuscript is of interest from the point of elucidating the self-regulatory mechanism of oxidative stress resistance in cartilage tissue. However, there are some concerns, and they are discussed below.

1. The authors added the fluid and measured oxidative stress level, redox status, and cell proliferation. The reviewer is curious whether the fluid has any effects on chondrocytic differentiation or not.

Response: We did not observe changes in chondrocyte morphology following post-operation knee fluid (POKF) exposure; chondrocytes maintained their typical rounded, polygonal morphology without showing any change toward a differentiated morphology. We are unsure whether 12 hours of KPOF exposure could be an appropriate time window to trigger any potential differentiation.

  1. The authors used the fluid and serum at the final concentration of 5% in cell culture experiments. The reviewer is curious how the authors decided the concentration?

Response: We carried out the experiments with a 5% concentration value since this value is normally used for bovine serum in culture media that require concentrations between 5 to 10%. We wanted to mimic a physiological situation where certainly the cells are in their best conditions. In our experiments, we also used the fluid and the serum at 10% concentration (data not shown), obtaining similar results to those with 5%. Moreover, these same values were also used in our past studies in which different cells had been treated with different sera of rheumatology and pulmonary patients [5-7].

  1. Also, the reviewer is curious in which ingredient(s) has inhibitory effect on oxidative stress. Did the author try to narrow down candidates such as heat denaturation against protein, and separation with centrifugal filtration against exosomes?

Response: The evolution of osteoarthritis causes the progressive deterioration of the joint, ultimately requiring a joint prosthesis implant to obtain joint recovery in the disease's advanced stages. To facilitate recovery, a drainage system is put in place during surgery to prevent blood from collecting in the joint after the postoperative bleeding. Previous studies have identified plasma components in this liquid (e.g., anti-inflammatory cytokines) capable of promoting anti-inflammatory processes and tissue repair [1-3]. However, so far, no studies have identified the exact constituents of the fluid and the components present in it. In this regard, although usually discarded, our work aimed to evaluate the potential protective effect of the fluid in oxidatively stressed chondrocytes, which is the study's novelty. In this view, we did not analyze the postoperative fluid content. Nonetheless, taking into account the reviewer's valuable suggestion, among the future objectives of this study, we intend to investigate the fluid composition and analyze the effect of different components, including potential exosomes or microvesicles, in consideration of the promising results of preclinical studies using mesenchymal stem cell (MSC)-derived exosomes [4].

References

  1. Wang-Saegusa, A., et al., Infiltration of plasma rich in growth factors for osteoarthritis of the knee short-term effects on function and quality of life. Archives of orthopaedic and trauma surgery, 2011. 131: p. 311-317.
  2. Filardo, G., et al., Platelet-rich plasma intra-articular knee injections for the treatment of degenerative cartilage lesions and osteoarthritis. Knee Surgery, Sports Traumatology, Arthroscopy, 2011. 19: p. 528-535.
  3. Baltzer, A., et al., Autologous conditioned serum (Orthokine) is an effective treatment for knee osteoarthritis. Osteoarthritis and cartilage, 2009. 17(2): p. 152-160.
  4. Kim, Y.G., J. Choi, and K. Kim, Mesenchymal Stem Cell‐Derived Exosomes for Effective Cartilage tissue repair and treatment of osteoarthritis. Biotechnology journal, 2020. 15(12): p. 2000082.
  5. Giordo, R., et al., Iloprost attenuates oxidative stress-dependent activation of collagen synthesis induced by sera from scleroderma patients in human pulmonary microvascular endothelial cells. Molecules, 2021. 26(16): p. 4729.
  6. Posadino, A.M., et al., NADPH-derived ROS generation drives fibrosis and endothelial-to-mesenchymal transition in systemic sclerosis: Potential cross talk with circulating miRNAs. Biomolecular Concepts, 2022. 13(1): p. 11-24.
  7. Ramli, I., et al., Low concentrations of Ambrosia maritima L. phenolic extract protect endothelial cells from oxidative cell death induced by H2O2 and sera from Crohn's disease patients. Journal of Ethnopharmacology, 2023. 300: p. 115722.

Round 2

Reviewer 2 Report

Comments and Suggestions for Authors

Please correct during proof reading:

line 92 "growth factors-associate action" write "associated"

line 255 and line 505: "H2O2-induced" write subscript

Reviewer 3 Report

Comments and Suggestions for Authors

In this revised manuscript, the authors seem to address the reviewers' comments adequately.